# Immune Response Modifications in the Genetic Forms of Parkinson’s Disease: What Do We Know?

**DOI:** 10.3390/ijms23073476

**Published:** 2022-03-23

**Authors:** Luca Magistrelli, Elena Contaldi, Francesca Vignaroli, Silvia Gallo, Federico Colombatto, Roberto Cantello, Cristoforo Comi

**Affiliations:** 1PhD Program in Clinical and Experimental Medicine and Medical Humanities, University of Insubria, 21100 Varese, Italy; magis.luca@gmail.com; 2Movement Disorders Centre, Neurology Unit, Department of Translational Medicine, University of Piemonte Orientale, 28100 Novara, Italy; contaldie@yahoo.it (E.C.); 20007636@studenti.uniupo.it (F.V.); 20032525@studenti.uniupo.it (S.G.); 20042514@studenti.uniupo.it (F.C.); roberto.cantello@med.uniupo.it (R.C.); 3PhD Program in Medical Sciences and Biotechnology, University of Piemonte Orientale, 28100 Novara, Italy

**Keywords:** Parkinson’s disease, immune system, alpha-synuclein

## Abstract

Parkinson’s disease (PD) is a common neurodegenerative disease characterized by loss of dopaminergic neurons in the *pars compacta* of the midbrain *substantia nigra*. PD pathophysiology is complex, multifactorial, and not fully understood yet. Nonetheless, recent data show that immune system hyperactivation with concomitant production of pro-inflammatory cytokines, both in the central nervous system (CNS) and the periphery, is a signature of idiopathic PD. About 5% of PD patients present an early onset with a determined genetic cause, with either autosomal dominant or recessive inheritance. The involvement of immunity in the genetic forms of PD has been a matter of interest in several recent studies. In this review, we will summarize the main findings of this new and promising field of research

## 1. Introduction

Parkinson’s disease (PD) is a common neurodegenerative disease with a prevalence of about 0.3% in the general population, increasing up to 3% in subjects over 80 years [1]. PD is clinically defined by the presence of bradykinesia, rest tremor, and rigidity which at the beginning are unilateral and spread contralaterally during the disease course. Patients also complain of several non-motor manifestations like hyposmia, constipation, depression, pain, and cognitive decline, which may precede by several years the onset of the classical motor phenotype and have an important impact on the quality of life of both patients and caregivers [2]. 

PD is characterized by loss of dopaminergic neurons in the *pars compacta* of the midbrain *substantia*
*nigra* with the accumulation of Lewy bodies in the surviving neurons, consisting in abnormal aggregates of proteins with alpha-synuclein (α-syn) representing the most expressed. The exact pathogenetic mechanisms leading to the neurodegenerative process are not yet fully understood but several factors must be taken into account with genetics and immunity playing relevant roles. 

Hyperactivation of immune cells is well documented, both in the periphery and in the central nervous system (CNS), probably due to alterations of the blood–brain barrier (BBB). Centrally, the activation of the microglia and astrocytes determine conversion to a detrimental phenotype leading to the neuronal loss, hence contributing to the worsening of symptoms [3]. This neurodegenerative process is further exacerbated by the infiltration, through the disrupted BBB, of peripheral circulating cells, like lymphocytes and monocytes, driven also by the central overexpression of α-syn [4]. 

Notwithstanding, the connection between CNS and the periphery is also mediated by the strict connection between the gut and the brain, with the so-called gut–brain axis. Through this bidirectional pathway, toxins and cytokines produced by a dysregulated gut microbioma can reach the CNS either directly through the vagus nerve, or indirectly with the blood circulation where they are released as a consequence of microbial translocation due to augmented intestinal permeability [5].

In the periphery, PD patients present a pro-inflammatory immune phenotype characterized by an imbalance between T helper Th1 and Th2 subpopulations (with a prevalence of the disruptive Th1 population) and a deficit in the T regulatory (Treg) cells which, in physiologic conditions, would counterbalance the detrimental immune activation. In addition, high concentrations of circulating pro-inflammatory cytokines, such as interferon-gamma (IFN-γ) and tumor necrosis factor-alpha (TNF-α), have been detected in PD patients [6]. Furthermore, T lymphocytes directly recognize specific α-syn epitopes and generate cytotoxic responses in PD patients [7]. Notably, recent data suggest that an impaired immune function may also be involved in the development of non-motor symptoms (like RBD and cognitive decline) [8,9] and motor fluctuations [10]. 

PD generally starts in the sixth/seventh decade, and patients with earlier onset, who often have a genetic cause, account for 3–7% of all cases [11]. Both autosomal dominant, recessive, and X-linked inheritance patterns have been described. Mutations and copy number variations (CNV) in the α-syn gene (*SNCA*) were the first genetic cause reported [12]. Subsequently, other mutations have been identified with *Parkin*, *PINK1*, and *DJ1* representing the most common recessive, and *SNCA* and leucine-rich repeat kinase 2 (*LRRK2)* the most frequent dominant, forms [11]. 

Overall, genetic forms of PD are characterized by an earlier age at onset, a higher burden of motor and non-motor symptoms, and a good response to advanced therapies like deep brain stimulation [13]. Besides causative genes, genetic risk factors, like mutations in the glucocerebrosidase gene (GBA), have been well characterized. When in heterozygosis, GBA mutations confer a five-fold increased risk of developing an early onset PD with predominant psychiatric comorbidities [14]. 

In this review, we discuss how gene mutations impact on phenotype and functioning of the immune system, and the implications that immune modifications may have on disease course. 

## 2. LRRK2

Mutations in *LRRK2* represent the most common form of monogenic PD with a dominant pattern of inheritance. Clinical presentation is quite similar to idiopathic PD (IPD) both in terms of age at onset and motor phenotype. The crosstalk between microglial LRRK2 and α-syn may be involved in disease progression: the toll-like receptor 2 (TLR2)-specific α-syn, released by dopaminergic neurons, activates neuro-inflammatory responses mediated by microglia via LRRK2, thus promoting further neurodegeneration. Accordingly, it was found that LRRK2 inhibition can counteract neuronal loss in murine *substantia nigra,* opening the question on a future utility of this therapeutic strategy [15]. LRRK2 is highly expressed in the lung, kidney, intestine, and in immune cells, where it plays an important role in inflammatory response regulation. The importance of this gene in immune regulation is testified by the finding that LRRK2 mRNA expression differs among immune cells. More in detail, while its expression is low or absent in T cells, higher levels can be detected in B lymphocytes [16]. Furthermore, implications in both cell differentiation and function have been hypothesized. Indeed, LRRK2 mRNA levels are lower in pre-B compared to mature B cells, where it is more expressed in the B2 subtype, and its inhibition determined an impairment of myeloid progenitors and myeloid cell maturation [16]. The studies concerning the regulation of function employ phorbol-12-myristate-13-acetate (PMA) or lipopolysaccharide (LPS) as pro-inflammatory stimuli. Accordingly, intra-nigral or systemic administration of LPS in PD murine models induces a central and peripheral pro-inflammatory status characterized by an increased release of damaging cytokines like TNF-α, IL-6 and IL-1, similar to what has been described in PD patients [17]. In this context, in LPS-induced inflammation, LRRK2 mRNA levels in B cells are downregulated, thus indicating a possible role of LRRK2 in the maintenance of the cellular resting state [16]. 

The G2019S mutation represents the most commonly described mutation in *LRRK2*. Park and colleagues tested whether it may have an impact on the immune system in transgenic mice. The authors observed that mice carrying this mutation had decreased numbers of monocytic and granulocytic progenitors in the bone marrow together with an increased proportion of circulating immature elements which block Th17 cells differentiation [18]. Th17 lymphocytes are known to produce interleukin (IL)-17, which promotes neutrophil recruitment and activation [19]. Accordingly, other lines of evidence reported that mice carrying G2019S mutations have a higher predisposition to fungal and bacterial infections [18]. The role of LRRK2 in host immune reactions was further confirmed by Gardet et al. [20], who showed that, in physiological conditions, LRRK2 expression is increased upon stimulation with IFN-γ, whereas protein deficiency due to *LRRK2* mutation leads to impairment of host defences. LRRK2 is also involved in the modulation of circulating cytokines. T and B lymphocytes from PD patients express high levels of LRRK2, which positively correlate with the production of pro-inflammatory cytokines like IFN-γ and TNF-α. This effect is not limited to lymphocytes but has also been confirmed in monocytes, which presented an even greater pro-inflammatory cytokine production [21]. 

High levels of LRRK2 can also be detected in myeloid cells like the CD14+CD16+ monocyte subgroup, neutrophils, and dendritic cells [22]. Although LRRK2 expression is low in resting cells, stimulation with IFN-γ consistently increases its expression as shown in pro-inflammatory M1 macrophages and activated CD14+CD16+ monocytes [22,23]. LRRK2 is a key regulator protein also in dendritic cells, which are antigen-presenting cells (APC), playing an important role in Treg maturation. In more detail, LRRK2 influences cell migration through the modulation of ORAI2, which cooperates with calcium channels in the regulation of calcium efflux, necessary for cell homeostasis [24]. 

Besides G2019S, the R1441G mutation is the second most frequent *LRRK2* mutation. Gillardon et colleagues showed that mice-derived microglia cells carrying this mutation displayed significantly increased levels of mRNA of pro-inflammatory cytokines, i.e., IL-1; IL-12; chemokine CC motif ligand 4 (CCL4); C-X-C Motif Chemokine Ligand 1 (CXCL1); and CCL3L1, broadening the immune regulation properties exerted by this gene [25]. 

In summary, several lines of evidence underline the crucial role of LRRK2 in the regulation of inflammatory mechanisms, both peripherally and centrally. Nonetheless, more work is needed to unravel, for example, the implications of specific *LRRK2* mutations in immune pathways. In this regard, a powerful approach could be represented by human-induced pluripotent stem cell (hiPSC) technology. A recent study by Panagiotakopoulou et al. [26], employing neurons and microglia derived from hiPSC, observed that the LRRK2 G2019S mutation, via the synergistic interaction with IFN-γ signaling, increased neuronal vulnerability to immune challenge. Through the analysis of molecular and cellular phenotypes deriving from genetic risk factors, researchers will hopefully be able to achieve novel insights for disease modeling and therapeutic intervention [27]. 

## 3. SNCA

*SNCA* was the first mutated gene detected in PD [12] and both point mutations and gene multiplications have been described. Patients carrying *SNCA* mutations present at a younger age at onset and with an early and severe non-motor symptoms constellation [28]. 

*SNCA* encodes for α-syn, a protein displaying multiple functions that are not yet fully understood. Recent evidence showing that α-syn is recognized by immune cells [7], has raised the interest over its relationship with the immune system. In vitro cultures of microglial cells from *SNCA* knock-out mice show different morphology and display higher baseline levels of activation markers (such as CD68 and β1 integrin) and cytokines secretion (TNFα and IL-6) upon LPS stimulation than wild-type (WT) mice. *SNCA* is also involved in immune cells’ maturation: *SNCA* knock-out mice have lower levels of both CD4+ and CD8+ cells. When analyzing different cell subtypes, these mice had a shift toward Th1 lymphocytes along with an impairment of Treg [29]. In vivo and in vitro data suggest that *SNCA* mutations preferentially mediate pro-inflammatory responses. Accordingly, SH-SY5Y cells carrying the A53T mutation had a more robust ability to activate co-cultured microglial cells, inducing the production of higher levels of IL-1, a pro-inflammatory cytokine [30]. These data were further confirmed by Gao [31] and La Vitola [32] who treated A53T mutated and WT mice with intraperitoneal LPS injection. A53T mice treated with LPS for 2.5 months presented a persistently hyperactivated neuro-inflammation state, as well as a progressive nigrostriatal neuronal loss with Lewy body-like inclusions in the nigral neurons and an accumulation of aggregated α-syn. Moreover, La Vitola showed that *SNCA* mutated mice had more pronounced cognitive deficits along with reduced astroglial markers, i.e., glial fibrillary acidic protein (GFAP), normally involved in glial-neurons communications and BBB integrity [32]. Interestingly, Roodveldt and colleagues showed that also two other mutations, A30P and E46K, had a robust capacity in inducing a pro-inflammatory response in microglia [33], which seemed to be more robust than the one induced by the A53T mutations, but these data need further confirmation. 

As highlighted in these studies, alterations in SNCA expression may mostly lead to functional changes in the microglia inflammatory phenotype. Nonetheless, knock-out animal models present several limitations and may not completely mimic the pathogenetic cascade related to genetic alterations of SNCA in humans. In this regard, an experiment by Haenseler et al. [34] using hiPSC-derived macrophages (which recapitulate many features of brain-resident microglia) from PD patients with the SNCA A53T mutation or SNCA triplications found in the latter elevated levels of α-syn and significantly compromised phagocytosis. Furthermore, CXCL1 and the pro-inflammatory cytokines IL-18 and IL-22 were upregulated in SNCA triplication macrophages versus controls, thus suggesting the dysregulation of cytokine production. However, further studies are warranted to understand the influence of endogenous SNCA expression on microglias’ immune functions in PD. 

## 4. Parkin

*Parkin* mutations, together with copy number variations, represent the most common autosomal recessive PD. Clinical presentation is characterized by a young age at onset (even in childhood) with a relatively slow progression and an initial good response to treatment [35]. 

*Parkin* (PRKN) encodes for an E3 ubiquitin ligase involved in the removal of damaged mitochondria [36]. Repeated intraperitoneal injections of low-dose LPS for 2, 3, or 6 months to either Parkin knock-out or WT mice showed that Parkin deficiency increased inflammatory-triggered dopaminergic degeneration [37]. Accordingly, knock-out mice also developed more severe motor deficits with a significantly slower average time to cross. The neuroinflammatory responses induced by LPS were evident in the olfactory bulb and midbrain, highlighting the susceptibility of these structures in the early phases of the disease. 

Parkin is also expressed in astrocytes, where it exerts anti-inflammatory activities by inhibiting NOD2, a cytosolic receptor involved in endoplasmic reticulum stress. In vitro studies showed that Parkin knock-out astrocytes displayed more pronounced endothelium stress with higher pro-inflammatory cytokines release and reduced levels of neurotrophic factors [38]. Notably, the replacement of Parkin was able to ameliorate inflammation in NOD2 positive astrocytes. 

## 5. PINK1

*PINK1* mutations represent the second most common cause of recessive PD. Besides the classical motor phenotype, patients complain of several psychiatric symptoms and signs [39]. In recent years, several studies have investigated the impact of mutated *PINK1* on immune mechanisms in PD. In 2016, Matheoud et al. provided in vivo and in vitro data in support of the role of PINK1 and Parkin in inhibition of mitochondrial antigen presentation (MitAP) and formation of mitochondria-derived vesicles (MDVs), on which MitAP depends [40]. Under cellular stress conditions (e.g., inflammatory state), loss of PINK1 activity triggers MitAP in dendritic cells and leads to the selection of peripheral mitochondrial antigen-specific T-cell populations thus driving a cytotoxic response and ultimately dopaminergic neuronal cell death. This hypothesis was further confirmed in an in vivo study, in which an intestinal infection with Gram-bacteria of PINK1 knock-out mice engaged MitAP and mitochondria-specific cytotoxic T-cells in the periphery and the brain. Intriguingly, the authors observed that in infected PINK1 knock-out mice the density of dopaminergic axonal varicosities in the striatum was markedly decreased and these mice displayed motor impairment that was reversed after L-DOPA treatment [41]. Taken together, these data support the role of PINK1 as a suppressor of the immune system, highlighting the connection between intestinal infection and immunity as key players in the pathophysiology of PD. 

Interestingly, both PINK1 and Parkin are part of the molecular pathway that modulates mitophagy [42]. 

## 6. PARKIN/PINK1 Interaction

PARKIN (PRKN) interacts with PINK1 in the elimination of damaged mitochondria thus controlling cellular energy metabolism [43]. In this context, Borsche M et al. [44] evaluated circulating levels of mitochondrial DNA (cmtDNA) in PD patients (with and without PRKN/PINK1 mutations) compared to healthy controls. Not only were cmtDNA levels significantly higher in patients than controls, but they were also well discriminated between PD patients carrying PRKN/PINK1 mutations and patients with IPD (area under the receiver operator characteristic curve = 0.81), thus indicating the higher degree of impaired mitophagy in PRKN/PINK1 associated PD. Notably, these patients also presented increased levels of IL-6 (a pro-inflammatory cytokine) which positively correlated with C-reactive protein (CRP), strengthening the concept of direct modulation of these genes on peripheral inflammation. Furthermore, in biallelic PRKN/PINK1 knock-out mice the Parkin or PINK1 deficiency led to cyclic GMPD-AMP synthase (CGAS)/ stimulator of interferon genes (STING)-dependent activation of pro-inflammatory response, associated with increased IL-6 levels which perpetuates the deleterious hyperactivation of the immune system [45].

## 7. DJ1

*DJ-1* represents about 1–2% of the monogenic forms of PD [46]. It has a recessive pattern of inheritance and is clinically characterized by an earlier age at onset with a motor spectrum similar to the idiopathic form. The encoded protein works as a redox sensor and may be implied in mitochondrial homeostasis.

To investigate whether *DJ1* deficiency may impact microglia activity, Trudler and colleagues exposed murine-derived microglial cells to an inflammatory stimulus with LPS. DJ1 knock-out cells had higher mitochondrial activation with consequent increased production of reactive oxygen species (ROS) and nitric oxide (NO). This detrimental scenario was further supported by the increased release of pro-inflammatory cytokines like IL-6, IL-1β along with a reduction of TNFα levels [47]. Interestingly, dopamine exposure determined a significant restoration of NO and ROS production through the activation of monoamine oxidase (MAO) enzymes. Treatment with MAO inhibitors led to a significant reduction of both NO and ROS levels (17 and 25%, respectively, from basal levels). 

Since human reactive astrocytes from PD patients overexpress DJ1 [48], Waak et al. analyzed murine-derived DJ-1 knock-out astrocytes to further investigate its role in the neurodegenerative process. The authors observed that these cells, after LPS injection, produced significantly higher levels of NO, IL-6, and Cyclooxygenase (COX)-2, which contributed to neuronal apoptosis, thus supporting the protective role of DJ1 against the neuro-inflammatory processes [49]. 

Besides microglial cells, the neurodegenerative process is also mediated by T lymphocytes that can reach the CNS through a disrupted BBB. DJ1 is also expressed in CD4+ T lymphocytes and mediates anti-oxidant activity through the downregulation of the Na+/H+ Exchanger 1 (NHE1), whose levels are increased in oxidative stress conditions [50]. Accordingly, as demonstrated by Zhou et al., murine-derived DJ1 negative CD4+ T cells had significantly higher levels of NHE1 transcripts and protein, with consequent increased ROS production [51]. 

This pro-inflammatory status detected in DJ1 deficiency may also be mediated by gut microbiome dysregulation, thus altering the so-called gut–brain axis whose dysfunction is widely involved in the pathophysiology of PD [5]. Notably, DJ1 knock-out mice presented a significantly higher production of fecal calprotectin and monocyte chemotactic protein 1 (MCP-1), along with both serum and fecal higher levels of malonate [52], which can mediate the degeneration of dopaminergic neurons [53]. Interestingly, DJ1-lacking mice displayed an alteration in the innate immune cells expressed with a significant reduction of CD45+ cells, indicating that DJ1 may also be involved in immune cells’ maturation. In this context, DJ1 has been implied in the development of Treg: lack of this protein determined a reduction in the Treg compartment and particularly induced Treg (iTreg), which showed impaired replicative and proliferative functions and increased susceptibility to cell death. This finding was probably explained by a significantly enhanced production of ROS in iTreg from DJ1 deficient mice [54]. 

## 8. GBA

Heterozygotic mutations in the glucocerebrosidase gene (GBA) gene, encoding the enzyme β-glucocerebrosidase (GCase), represent the principal risk factors for PD. These patients present a greater risk for dementia and psychiatric symptoms [55]. Furthermore, a correlation between residual enzymatic activity and clinical severity of GBA mutations has been reported [56]. Few studies have investigated the involvement of peripheral immune cells in PD-GBA. A case-control study [4] analyzed the difference in plasmatic proteins’ expression between PD with and without the GBA mutation and showed that mutated PD patients had higher plasma levels of monocytes’ inflammatory mediators, especially IL-8 [57]. Moreover, higher IL-8 plasma levels seemed to correlate with worse cognitive performances. Another study reported significant expression changes of 26 genes in the peripheral blood leukocytes of PD-GBA compared with healthy controls [58]. Most of these genes were down-regulated, specifically those related to B cell function (specifically the B cell receptor signaling pathway), and a correlation with disease duration was observed. Peripheral blood cells may represent a suitable source for GCase measurement: a study by Alcalay et al. identified a significant decrease in GCase activity in whole blood samples from idiopathic PD patients [59]. Nonetheless, heterogeneous results using peripheral mononuclear cells have been reported [60]. Atashrazm et al. observed that, in PD patients, GCase activity was significantly reduced in monocytes (but not in lymphocytes) compared with controls, even when GBA mutation carriers were excluded [61]. The selective decrease in monocytes is a relevant finding, as they represent a small percentage of total leukocytes and reduced GCase activity in heterogeneous cell populations could be difficult to detect [61]. Furthermore, comparative transcriptome analysis in monocyte-derived macrophages of four asymptomatic GBA mutation carriers, four controls, and five PD-GBA, revealed in the latter group the deregulation of genes involved in the immune response, mostly related to monocyte and neutrophil chemotaxis, myeloid leukocyte migration, and cellular response to chemokines [62]. Several studies reported monocyte dysfunction with impaired responsiveness to stimulation as well [63,64]. Indeed, in Gaucher disease (GD), a lysosomal storage disorder caused by homozygous or compound heterozygous mutations in GBA, monocyte activation markers have been validated as biomarkers for monitoring therapeutic response [65]. Based on these findings, a recent study by Galper et al. explored whether ferritin, cluster of differentiation (CD)162, CC Chemokine Ligand (CCL)18, and chitotriosidase (immune biomarkers typically associated with GD) were also altered in GBA carriers with or without PD, but these plasma biomarkers were not relevant for the stratification of PD risk in carriers of heterozygous GBA pathogenic variants [66]. Thaler et al. did not detect significant differences in the plasmatic and cerebrospinal fluid (CSF) levels of several inflammatory cytokines among PD-GBA, PD-LRRK2, and idiopathic PD patients [67]. In conclusion, there are certain discrepancies in the studies exploring the relationship between GBA-PD and the alteration of immune networks: future research assessing the exact role of inflammation in this subset of patients will hopefully aid targeted therapeutic interventions.

The main findings are summarized in Figure 1.

## 9. Conclusions

The understanding of the complex relationships between PD genes and immunity is at its first steps. The findings are intriguing but still very preliminary. Genetic PD is rare and conducting a longitudinal study, which is the best approach to address causal relationships between immunity and neurodegeneration, will require a great collaborative effort. In the meantime, answers are coming from animal models, whose findings have then to be confirmed in patients. Novel research approaches, such as disease modeling with hiPSC, may add great value to this respect. Co-culturing of dopaminergic neurons, microglia, and immune cells may become a precious tool to generate hypotheses to be validated in future longitudinal studies. Among the points of interest of this line of research, there is certainly a more traceable molecular signature of genetic compared to idiopathic PD that should ultimately help in better define how, when, and where, the immune system intersects with the neurodegenerative process, and consequently open the way to a personalized therapeutic approach. 

## Figures and Tables

**Figure 1 ijms-23-03476-f001:**
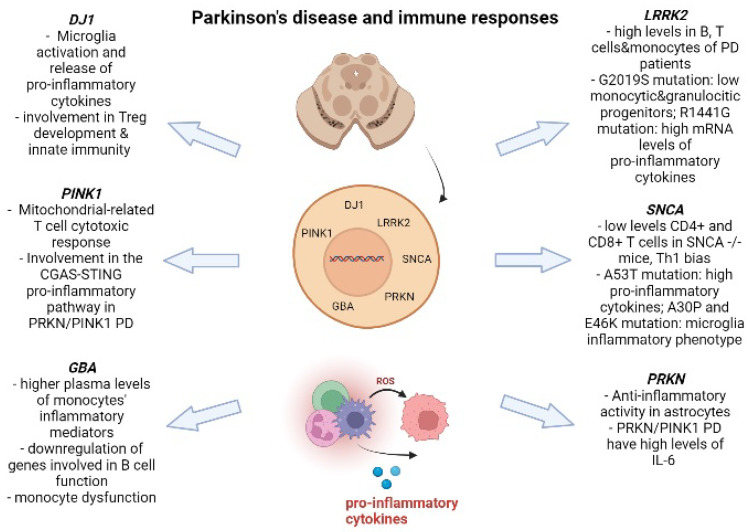
Interactions between PD genetic mutations and immune system. Created with www.biorender.com.

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
