# Peer review of "Immune Response Modifications in the Genetic Forms of Parkinson’s Disease: What Do We Know?"

_ijms, 2022, doi:10.3390/ijms23073476_

Round 1

Reviewer 1 Report

The work is of interest, there are some minor comments/questions:

The section about parkin involves some data concerning PINK1 as well, but the section on PINK1 can be found only later, involving some comments on PRKN/PINK1 knockout mice as well. The interaction and mutual effects of PRKN and PINK1 may be discussed in a separate paragraph.

line 34: “with the accumulation in the surviving neurons of Lewy bodies” accumulation of what?

The effect of LPS is often mentioned concerning either LRRK2, SNCA, parkin or DJ1 mutations. It would be interesting to know the general effect of LPS in idiopathic PD for the sake of comparison.

A figure or scheme highlighting the connection of the major proteins with the immune system would help.

Author Response

Reviewer 1

- The section about parkin involves some data concerning PINK1 as well, but the section on PINK1 can be found only later, involving some comments on PRKN/PINK1 knockout mice as well. The interaction and mutual effects of PRKN and PINK1 may be discussed in a separate paragraph.

AR: we thank the reviewer for the suggestion. Accordingly, the paragraphs describing PRKN and PINK1 are now closer and a specific section dealing with their interaction as been added

- line 34: “with the accumulation in the surviving neurons of Lewy bodies” accumulation of what?

AR: according to the reviewer suggestions the concept has been better explained: “PD is characterized by loss of dopaminergic neurons in the pars compacta of the midbrain substantia nigra with the accumulation of Lewy bodies in the surviving neurons, consisting in abnormal aggregates of proteins with alpha-synuclein (α-syn) representing the most expressed”.

- The effect of LPS is often mentioned concerning either LRRK2, SNCA, parkin or DJ1 mutations. It would be interesting to know the general effect of LPS in idiopathic PD for the sake of comparison.

AR: we thank the reviewer for the suggestion. Accordingly, this concept has been explained in the text explaining as the LPS induced PD murine model better explain the pro-inflammatory status which has been also confirmed in both patients and genetic forms of PD: “The studies concerning the regulation of function employ phorbol-12-myristate-13-acetate, PMA, or lipopolysaccharide, LPS as pro-inflammatory stimuli. Accordingly, intra-nigral or systemic administration of LPS in PD murine models induce a central and peripheral pro-inflammatory status characterized by an increased release of damaging cytokines like TNF-α, IL-6 and IL-1, similarly to what has been described in PD patients [15]”.

- A figure or scheme highlighting the connection of the major proteins with the immune system would help.

AR: accordingly, a summary figure was added

Reviewer 2 Report

Magistrelli et al. review the immune response in different genetic forms of Parkinson’s disease (PD). They first provide a succinct general introduction on PD and inflammation. The rest of the review describes the main alterations in central and peripheral immune responses observed in different genetic forms of Parkinson’s disease. Overall, the review is interesting. It is well-written, timely and provides a succinct overview of immunity in the context of genetic PD.

I have some minor comments that could improve the review. My main concern is that there is not any image or schematic summary of the topics discussed, and including one would be of great help for the readers.

Also, infiltration of peripheral cells and the gut-brain axis, which are only discussed in the context of DJ1 PD, should also be discussed in other forms of PD, as they are not specific for DJ1. For example, it has been shown that dopaminergic neuronal loss is dependent on monocyte infiltration into the CNS in a mouse model that overexpresses human α-synuclein (Harms et al., 2018).

Author Response

Reviewer 2

- I have some minor comments that could improve the review. My main concern is that there is not any image or schematic summary of the topics discussed, and including one would be of great help for the readers.

AR: accordingly, a summary figure was added

- Also, infiltration of peripheral cells and the gut-brain axis, which are only discussed in the context of DJ1 PD, should also be discussed in other forms of PD, as they are not specific for DJ1. For example, it has been shown that dopaminergic neuronal loss is dependent on monocyte infiltration into the CNS in a mouse model that overexpresses human α-synuclein (Harms et al., 2018).

AR: we thank the reviewer for this suggestion. Accordingly, this aspect has been better explained in the introduction section: “This neurodegenerative process is further enhanced by the infiltration, through the disrupted BBB, of peripheral circulating cells, like lymphocytes and monocytes, driven also by the central over expression of α-syn [4].  Notwithstanding, the connection between CNS and the periphery is also mediated by the strict connection between the gut and the brain with the so-called gut-brain axis. Through this bidirectional pathway, toxins and cytokines produced by a dysregulated gut microbioma can reach the CNS directly through the vagus nerve or indirectly with the blood circulation where they are released as a consequence of microbial translocation due to augmented intestinal permeability [5]”
